# OmniTokenizer: A Joint Image-Video Tokenizer for Visual Generation

**Junke Wang**[1,2], **Yi Jiang**[3,♠], **Zehuan Yuan**[3], **Binyue Peng**[3], **Zuxuan Wu**[1,2†,♠], **Yu-Gang Jiang**[1,2]

[1]Shanghai Key Lab of Intell. Info. Processing, School of CS, Fudan University
[2]Shanghai Collaborative Innovation Center on Intelligent Visual Computing, [3]Bytedance Inc.

Code available at https://github.com/FoundationVision/OmniTokenizer.

## Abstract

Tokenizer, serving as a translator to map the intricate visual data into a compact latent space, lies at the core of visual generative models. Based on the finding that existing tokenizers are tailored to image or video inputs, this paper presents OmniTokenizer, a transformer-based tokenizer for joint image and video tokenization. OmniTokenizer is designed with a spatial-temporal decoupled architecture, which integrates window and causal attention for spatial and temporal modeling. To exploit the complementary nature of image and video data, we further propose a progressive training strategy, where OmniTokenizer is first trained on image data on a fixed resolution to develop the spatial encoding capacity and then jointly trained on image and video data on multiple resolutions to learn the temporal dynamics. OmniTokenizer, for the first time, handles both image and video inputs within a unified framework and proves the possibility of realizing their synergy. Extensive experiments demonstrate that OmniTokenizer achieves state-of-the-art (SOTA) reconstruction performance on various image and video datasets, *e.g.*, 1.11 reconstruction FID on ImageNet and 42 reconstruction FVD on UCF-101, beating the previous SOTA methods by 13% and 26%, respectively. Additionally, we also show that when integrated with OmniTokenizer, both language model-based approaches and diffusion models can realize advanced visual synthesis performance, underscoring the superiority and versatility of our method.

## 1 Introduction

The development of generative models [25, 54, 14, 17, 10, 41] has been one of the most exhilarating developments in artificial intelligence, offering the potential to revolutionize the way we generate visual content. In recent years, visual generation approaches have emerged as two dominant paradigms: language model-based methods [54, 12, 66, 48] and diffusion models [17, 45]. The former exploits the superior sequence modeling capability of language models (LMs) [36, 37, 52] for visual generation by formulating it as a next-token prediction process, while the latter gradually transforms noise into coherent visual structures through a carefully crafted reverse diffusion process.

Core to both approaches is the *tokenizer*, which translates visual signals into latent representations, with LM tokenizers, also known as VQVAE, discretizing inputs into sequences of latent codes [12, 64, 66, 27], and diffusion tokenizers, *i.e.*, VAE, modeling their probability distributions within a latent space [25, 41]. Analogous to the role of the lexicon in a written language, tokenizers for visual synthesis dictate the upper bound of the generative models, thus attracting increasing attention in the community [12, 63, 19].

---

♠: project leaders, †: corresponding author.

Existing tokenizers are designed specifically for either image [12, 64] or video [63, 13, 66] inputs, resulting in inherent limitations regarding their application flexibility and data scalability for the following generative models. Although MAGVITv2 [67] have explored causal 3D convolution to process both modalities, they still have to train separate models for the image and video data, without achieving the synergy between them. This work highlights the critical need for a joint image-video tokenizer with two primary considerations: firstly, a joint image-video tokenizer enables joint learning from image and video data [59], which mitigates the scarcity of data in a single modality (particularly video data) and facilitates the tokenizer to learn more general representations. In addition, a unified tokenization framework inherently enjoys better versatility and scalability. For instance, its performance can be improved by incorporating the data from either modality for training. This further promotes the efficacy of generative models tailored to image or video generation.

With this in mind, we present OmniTokenizer, a transformer-based tokenizer for joint image-video tokenization. As intuitive as it may seem, the simple unification of image and video data could not lead to the reciprocal effects between both modalities. To address this challenge, we turn to a spatial-temporal decoupled architecture [2], where window attention [28] is employed in the spatial dimension owing to its local aggregation capacity and efficiency, and causal attention is used in the temporal dimension to capture the motion in videos and ensure temporal coherence. Complementing the model design, we introduce a progressive training strategy that begins with image pretraining on a fixed resolution to establish a fundamental understanding of static visual information. After this, we integrate video data for joint training on variable resolutions to capture the dynamics in more complex scenes. The progressive training strategy allows our method to bridge the gap between disparate forms of visual input and capitalize on the rich spectrum of visual data.

To empirically validate the effectiveness of the proposed method, we separately implement the LM and diffusion tokenizers, *i.e.*, OmniTokenizer-VQVAE and OmniTokenizer-VAE, and conduct experiments on a wide range of datasets including ImageNet [9], CelebA-HQ [21], FFHQ [22], UCF-101 [46], Kinetics-600 [6], *etc*. The results demonstrate our model outperforms existing methods in terms of reconstruction FID on both image datasets (*e.g.*, 1.11 rFID for OmniTokenizer-VQVAE and 0.69 rFID for OmniTokenizer-VAE on ImageNet) and video datasets (*e.g.*, 42 rFVD for OmniTokenizer-VQVAE and 23 rFVD for OmniTokenizer-VAE on UCF-101). In addition, employing our approach for tokenization, we also show that both language model-based generative models and diffusion models could achieve competitive results on class-conditional, unconditional generation, and frame prediction tasks.

In summary, our work makes the following key contributions:

- We introduce OmniTokenizer, a transformer-based tokenizer for joint image and video tokenization. For the first time, OmniTokenizer employs a shared framework and weight to handle both types of visual data.

- We propose a progressive training strategy that begins with image pre-training at a fixed resolution and then transits to image-video joint training at multiple resolutions. Such an approach capitalizes on the synergies between image and video data, facilitating OmniTokenizer to achieve better performance than solo image or video training.

- We conduct extensive experiments across various datasets like ImageNet, CelebA-HQ, FFHQ, UCF-101, and Kinetics-600. The results showcase the state-of-the-art reconstruction performance of OmniTokenizer on both image and video datasets. Furthermore, equipped with OmniTokenizer, both language model-based generative models and diffusion models could achieve superior generation results.

## 2 Related Work

### 2.1 Language Models for Visual Generation

Language models have emerged as powerful contenders in the visual generation field, drawing inspiration from their unparalleled success in natural language processing [36, 37, 51, 52] and visual understanding [11, 5, 30, 58, 57]. These methods [12, 7, 13, 66] recast visual synthesis as a sequence prediction problem, similar to constructing sentences in human language.

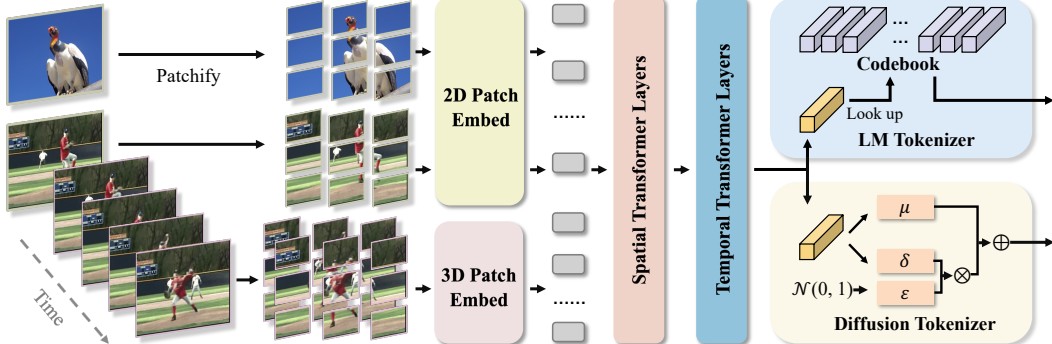

Figure 1: Architecture of OmniTokenizer, which consists of patch embedding layers, and separate spatial-temporal attention blocks. To obtain the latent representations, OmniTokenizer-VQVAE looks up a codebook to quantize the encoder embeddings, while OmniTokenizer-VAE samples from a Gaussian distribution. We omit the decoder and only show the tokenization process.

Depending on whether the tokens are predicted sequentially or in parallel, LM-based methods can be further categorized into autoregressive models [12, 65] and non-autoregressive models [7, 67]. Autoregressive (AR) models have been the initial foray into visual generation, utilizing the inherent sequential nature of language models to generate images [64, 65] and videos [63, 13] in a step-wise fashion. These models, such as DALL-E [39] and its preceding variants, typically work by predicting one token at a time and are characterized by their high-quality outputs and precise control over the generation process. VAR[48]redefines the autoregressive learning framework on images as coarse-to-fine "next-scale prediction" paradigm. Non-autoregressive (Non-AR) models, on the other hand, have been developed to allow for a faster generation process by predicting multiple tokens independently and in parallel. Models like MaskGIT [7] leverage this parallelism to significantly reduce generation time while maintaining high fidelity in synthesized images. The non-AR approaches have also demonstrated promise in video generation, featured by MAGVIT series [66, 67]. Both AR and non-AR methods have significantly advanced the field of visual generation, offering novel methods to synthesize high-quality images and videos.

## 2.2 Diffusion Models for Visual Generation

Diffusion models [17, 33, 3, 62, 61] represent an alternative avenue for visual generation, benefiting from their probabilistic nature that iteratively denoise a random signal into structured images or videos. These models stand out for their flexibility in generating visual outputs that not only exhibit coherent global structures but are also rich with intricate textures [32, 34]. Unlike language models that discretize visual inputs as latent codes, diffusion models directly generate visual samples in continuous pixel space [45, 10]. While effective, this approach demands significant computational resources given the high dimensionality of visual data.

The advent of latent diffusion models (LDMs) [41] seeks to mitigate these issues by compressing the high-dimensional visual data into latent space with a pretrained Variational Autoencoder (VAE) [25, 41]. LDM preserves the desirable properties of pixel-space diffusion models, such as high-quality image synthesis and the ability to incorporate conditional information, while drastically reducing the training and sampling overhead. After that, the rise of LDMs [71, 35, 34, 29] continues to push visual generation toward higher quality, larger resolution, and more complex scenes.

## 3 Methodology

### 3.1 Joint Image and Video Tokenization

We aim to enable image and video tokenization in a unified framework and achieve mutual benefits between them. To accomplish this, we employ a transformer-based architecture with decoupled spatial and temporal blocks (Sec. 3.1.1). Complementing this, we also propose a progressive training strategy consisting of two consecutive stages to learn the visual encoding in an incremental way (Sec. 3.1.2). The overall framework of our method is illustrated in Figure 1.

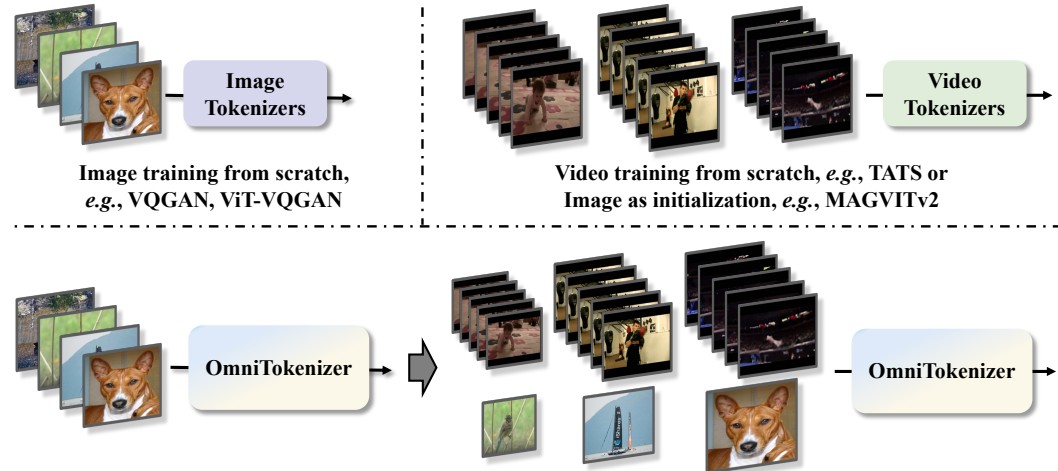

Figure 2: Illustration of the proposed progressive training paradigm. With this, OmniTokenizer could tokenize both image and video inputs with the same architecture and weight.

### 3.1.1 Space-Time Transformer

**Patchify.** Given a visual input $x \in \mathbb{R}^{(1+T) \times H \times W \times 3}$, where $(1 + T)$ is the number of frames ($T = 0$ for image) and $H \times W$ denotes the spatial resolution, we always process the first frame $x_0 \in \mathbb{R}^{1 \times H \times W \times 3}$ and following frames $x_{1:T} \in \mathbb{R}^{T \times H \times W \times 3}$ separately for the joint encoding of videos and static images [67]. Specifically, both $x_0$ and $x_{1:T}$ are split into non-overlapping patches, with a patch size of $p \times p$ and $t \times p \times p$, respectively. After that, we project the image and video patches with two linear layers, obtaining the patch embeddings $e_0 \in \mathbb{R}^{L_1 \times c}$ and $e_{1:T} \in \mathbb{R}^{L_2 \times c}$, where $L_1 = \frac{H}{p} \times \frac{W}{p}$ and $L_2 = \frac{T}{t} \times \frac{H}{p} \times \frac{W}{p}$. $e_0$ and $e_{1:T}$ are then concatenated along the sequence dimension as the spatial-temporal embedding $e$. In this way, we compress the input resolution from $(1 + T) \times H \times W$ to $(1 + \frac{T}{t}) \times \frac{H}{p} \times \frac{W}{p}$.

**Encoder and Decoder.** To have better compatibility with image and video inputs, we adopt a spatial-temporal factorized encoder consisting of separate spatial and temporal blocks. In the spatial dimension, window attention [28] is employed as it exhibits superior local aggregation capability and efficiency. While in the temporal dimension, we use causal attention to align with the autoregressive visual generation in the second stage. Next, the latent code $z$ could be obtained by looking up a codebook [54] for LM tokenizer (*i.e.*, quantization in VQVAE), or sampling from a Gaussian distribution for diffusion tokenizer.

The architecture of the decoder is symmetric with the encoder. Finally, we map the spatial-temporal tokens to the pixel space with two linear projection layers without any activation function.

### 3.1.2 Progressive Training

Unlike existing image tokenizers that conduct training on image data only [12, 64] or video tokenizers that train with image counterparts as intialization [66, 67]. We leverage a progressive training paradigm that involves two consecutive stages of VQ training to facilitate spatial-temporal representation learning of our LM tokenizer OmniTokenizer-VQVAE. After this, it could be fine-tuned as a diffusion tokenizer, OmniTokenizer-VAE, with KL fine-tuning.

**Two-stage VQ Training**, as depicted in Figure 2, aims to learn the visual reconstruction with the discrete latent codes. It includes two stages, the initial stage focuses on fixed-resolution image data to lay a foundation for spatial understanding. Building upon this, the second stage introduces video data to learn the modeling of temporal dynamics alongside static image features. This image-video joint training stage is critical for the model to learn a universal embedding that accurately captures both the spatial intricacies of individual frames and the temporal relationships of sequential video data.

During both stages, the model is trained with vector-quantization objective:

$$\mathcal{L}_{VQ} = \lambda_1 ||\text{sg}[E(e)] - z_q||_2^2 + \lambda_2 ||E(e) - \text{sg}[z_q]||_2^2, \tag{1}$$

Table 1: Reconstruction FID on ImageNet validation split, CelebA-HQ, and FFHQ. $^*$ denotes models trained with Gumbel-Softmax reparameterization [39]. For our method, the results that are jointly trained with UCF-101 are reported.

| Method | Dataset | Lat. shape | Codebook | rFID |
|---|---|---|---|---|
| ViT-VQGAN [64] | CelebAHQ | $32 \times 32$ | 8192 | 4.66 |
| Ours-VQVAE | CelebAHQ | $32 \times 32$ | 8192 | **1.93** |
| ViT-VQGAN [64] | FFHQ | $32 \times 32$ | 8192 | 3.13 |
| Ours-VQVAE | FFHQ | $32 \times 32$ | 8192 | **1.91** |
| DALL-E [39] | ImageNet | $32 \times 32$ | 8192 | 32.01 |
| VQGAN$^*$ [12] | ImageNet | $32 \times 32$ | 8192 | 1.49 |
| ViT-VQGAN [64] | ImageNet | $32 \times 32$ | 8192 | 1.28 |
| Ours-VQVAE | ImageNet | $32 \times 32$ | 8192 | **1.11** |
| Ours-VAE | ImageNet | $32 \times 32$ | $\infty$ | **0.69** |

Table 2: Reconstruction FVD on UCF-101 and Moments-in-Time val. split. $^*$ denotes training image tokenizer with video loss.

| Method | Type | UCF | MiT |
|---|---|---|---|
| MaskGIT [7] | Img | 240 | - |
| VQGAN [12] | Img | 299 | 306 |
| ViT-VQGAN [64] | Img | - | 167 |
| ViT-VQGAN$^*$ [64] | Img | - | 173 |
| CViViT [55] | Vid | - | 66 |
| TATS [13] | Vid | 162 | - |
| MAGVIT [66] | Vid | 58 | - |
| Ours-VQVAE | Joint | **42** | **20** |
| Ours-VAE | Joint | **23** | **13** |

where sg denotes the stop-gradient operation, $\lambda_1$ and $\lambda_2$ are the balancing hyperparameters, $E$ and $z_q$ represent the encoder of OmniTokenizer and codebook vectors, respectively. Factorized codes and $l_2$-normalized codes [64] are also used to boost the codebook usage.

**KL fine-tuning.** After the VQ training, we further fine-tune our model as a diffusion tokenizer (*i.e.*, OmniTokenizer-VAE) by replacing the above $\mathcal{L}_{VQ}$ with Kullback-Leibler (KL) loss:

$$\mathcal{L}_{KL} = \lambda_3 D_{KL}(Q(z|x)||P(z)), \tag{2}$$

where $P(z)$ is Gaussian distribution, $Q(z|x)$ represents the inferred posterior configurations of the latent code given the observed input.

Besides $\mathcal{L}_{VQ}$ or $\mathcal{L}_{KL}$, both VQ training and KL fine-tuning also employs $L_2$ reconstruction loss $\mathcal{L}_{recon}$ and GAN loss $\mathcal{L}_{GAN}$.

## 3.2 Visual Generation

As mentioned in Sec. 3.1.2, after the progressive training and KL fine-tuning, we can obtain two tokenizers: OmniTokenizer-VQVAE and OmniTokenizer-VAE which separately encode the visual inputs into latent codes in a discrete codebook or the continuous latent space. With this, we further train language models or diffusion models for visual generation.

**Language models-based generation approaches** formulate visual synthesis as a token prediction problem. Specifically, after OmniTokenizer-VQVAE tokenizes image or video inputs into a sequence of discrete latent codes, we first flatten them in the raster order [8, 12] to obtain the code indices $y$. Then a transformer language model [36] is trained to maximize the log-likelihood between the predicted tokens $\hat{y}$ and the target tokens $y$ with cross-entropy loss:

$$\text{maximize} \sum_{i=1}^{L} \log P(\hat{y}_i|c, y_{1:i-1}; \theta). \tag{3}$$

where $c$ represents the condition (*e.g.*, label for class-conditional image and video generation), $\theta$ is the learnable parameters of the language model, P and $L$ denote the softmax probability and the length of $y$. During inference, we predict each token according to the model likelihood.

**Latent diffusion models** (LDMs) [41] perform diffusion process in the latent space to enable high-quality image synthesis with improved computational efficiency. Specifically, with the 2D latent representation from OmniTokenizer-VAE, the diffusion process gradually applies Gaussian noise to the latent code to generate a perturbed sample, while the denoising process trains a diffusion model to predict the noise that has been added. During inference, the well-trained diffusion model could generate a coherent visual sample from the noise by iteratively reversing the noising process.

## 4 Experiments

**Datasets.** We evaluate the visual tokenization performance of OmniTokenizer on both image and video datasets, including ImageNet [9], CelebA-HQ [21], FFHQ [22], Kinetics [23, 6], UCF-101 [46],

Table 3: Comparions of class-conditional results on ImageNet 256×256 using language models. "↓" ("↑") indicates lower (higher) is better. Metrics include Fréchet inception distance (FID) and inception score (IS). NAR and AR: non-autoregressive and autoregressive. *: taken from MaskGIT [7].

| Type | Method | #Param | FID↓ | IS↑ |
|---|---|---|---|---|
| AR | VQGAN* [12] | 227M | 18.65 | 80.4 |
| AR | RQ-Transformer [26] | 488M | 15.72 | 86.8 |
| AR | Ours | 227M | **10.13** | **94.5** |
| AR | VQVAE-2* [40] | 13.5B | 31.11 | ∼45 |
| AR | VQGAN [12] | 1.4B | 15.78 | 74.3 |
| AR | RQ-Transformer [26] | 821M | 13.11 | 104.3 |
| AR | ViT-VQGAN [64] | 650M | 8.81 | 110.8 |
| AR | Ours | 650M | **7.45** | **146.7** |

Table 4: Comparions of class-conditional generation results on UCF-101 and frame prediction results on Kinetics-600. Fréchet video distance (FVD) is reported.

| Type | Method | #Param | FVD↓ | |
|---|---|---|---|---|
| | | | UCF | K600 |
| NAR | Phenaki [55] | 227M | - | 36.4 |
| NAR | MAGVIT [66] | 306M | 76 | 9.9 |
| NAR | MAGVITv2 [67] | 307M | 58 | 4.3 |
| AR | LVT [38] | 50M | - | 224.7 |
| AR | ViTrans [60] | 373M | - | 170.0 |
| AR | CogVideo [19] | 9.4B | 626 | 109.2 |
| AR | ViVQVAE [56] | NA | - | 64.3 |
| AR | TATS [13] | 321M | 332 | - |
| AR | Ours | 227M | 314 | 34.2 |
| AR | Ours | 650M | **191** | **32.9** |

Moments-in-Time (MiT) [31], and Something-Something v2 (SSV2) [15]. We adopt a subset of the above datasets for visual generation to compare with previous works [12, 64, 55, 13].

**Implementation Details.** OmniTokenizer adopts a decoupled spatial-temporal architecture consisting of 4 window attention-based spatial layers (window size = 8) and 4 causal attention-based temporal layers. The hidden dimension is 512 and the latent dimension is 8, following ViT-VQGAN [64]. $\lambda_1$, $\lambda_2$, and $\lambda_3$ are set to 1, 1, 1e-6, respectively. As mentioned in Sec. 3.1.2, the training of OmniTokenizer follows a progressive training strategy, where both stages last 500K iterations. The learning rate is warmed up to 1e-3 and decayed to 0 using a cosine scheduler. Adam [24] is employed for optimization ($\beta 1 = 0.9$ and $\beta 2 = 0.99$). During the image training stage, we train the model with a fixed image resolution of 256×256. For the joint training stage, we forward the model with image and video data iteratively, with the video sequence length being 17 frames. The spatial resolutions are randomly chosen from 128, 192, 256, 320, and 384 [49]. Only random horizontal flip is adopted for data augmentation. We train our model using 8 NVIDIA A100 GPUs for 2 weeks. Unless otherwise stated, the results reported in this paper are jointly trained on ImageNet and UCF-101.

We try both the language models and diffusion models for visual generation with OmniTokenizer as the tokenizer. The configuration for the language model follows VQGAN [12], and for a fair comparison with previous methods, we also scale up the model size by increasing the hidden dimension to 1535, following ViT-VQGAN [64]. The training of image and video diffusion transformers follows DiT [34] and Latte [29], respectively.

## 4.1 Visual Tokenization

We first evaluate the visual tokenization capability of OmniTokenizer on ImageNet and two high-quality face datasets, CelebA-HQ and FFHQ. Reconstruction FID is used following the previous methods [12, 64]. We can observe from Table 1 that with the same compression rate and codebook size, OmniTokenizer outperforms existing methods by a large margin on all these datasets. Especially, OmniTokenizer-VQVAE achieves 1.11 FID on ImageNet, beating ViT-VQGAN, the previous state-of-the-art method by 13%. When fine-tuned as OmniTokenizer-VAE, the FID is further reduced to 0.69. We hypothesize the improved performance is because KL training provides smoother gradients than VQ training and avoids loss of information in the quantization process.

In addition, we also conduct video reconstruction experiments and report the results in Table 2. We can see that on both UCF-101 and Moments-in-Time datasets, OmniTokenizer achieves the best results. The video reconstruction results on more datasets can be found in the ablation study.

## 4.2 Visual Generation with AutoRegressive Transformers

Using OmniTokenizer-VQVAE for tokenization, we train language models to predict latent code indices in the codebook in an autoregressive manner for image and video synthesis. The class-conditional 256×256 generation results on ImageNet, presented in Table 3, demonstrate that our

Table 5: Class-conditional results on ImageNet 256×256 using GAN and diffusion models.

| Method | FID↓ | IS↑ | Prec↑ | Rec↑ |
|---|---|---|---|---|
| BigGAN [4] | 6.95 | 171.4 | 0.87 | 0.28 |
| StyleGAN-XL [42] | 2.30 | 265.12 | 0.78 | 0.53 |
| ADM [10] | 10.94 | 100.98 | 0.69 | 0.63 |
| LDM-4 | 10.56 | 103.49 | 0.71 | 0.62 |
| CDM [18] | 4.88 | 158.71 | - | - |
| DiT-XL/2 [34] | 9.62 | 121.50 | 0.67 | **0.67** |
| DiT-XL/2-CFG [34] | **2.27** | **278.24** | 0.83 | 0.57 |
| Ours-DiT-XL/2 | 12.25 | 109.94 | 0.73 | 0.64 |
| Ours-DiT-XL/2-CFG | 3.48 | 244.23 | **0.89** | 0.52 |

Table 6: Comparisons of unconditional results on UCF-101 256×256 using GAN and diffusion models.

| Method | Lat. Comp. | FVD↓ |
|---|---|---|
| MoCoGAN [53] | - | 2886.9 |
| VideoGPT [63] | $4 \times 4 \times 4$ | 2880.6 |
| MoCoGAN-HD [50] | - | 1729.6 |
| DIGAN [69] | - | 1630.2 |
| StyleGAN-V [44] | - | 1431.0 |
| PVDM [68] | $1 \times 4 \times 4$ | 1141.9 |
| MoStGAN-V [43] | - | 1380.3 |
| Latte [29] | $1 \times 8 \times 8$ | 478.0 |
| Ours-Latte | $4 \times 8 \times 8$ | **209.2** |

model surpasses existing autoregressive image generation methods with significant margins. Remarkably, with a model comprising only 227M parameters, we achieve 10.13 FID and 94.5 IS, outperforming VQGAN [12] by 32% and 25%, respectively. Upon scaling up to a larger model with 650M parameters, the FID is further reduced to 7.45.

In the domain of video generation, as illustrated in Table 4, our model beats the previous state-of-the-art autoregressive model, TATS [13] for class-conditional video generation on UCF-101 with much lower FVD (283 *v.s.* 314). Moreover, for frame prediction tasks on the Kinetics-600 dataset, our model not only achieves the best performance compared to other autoregressive models but also surpasses Phenaki [55], a non-autoregressive method.

### 4.3 Visual Generation with Diffusion Models

In parallel to language model-based methods, diffusion model [17, 45, 10], especially latent diffusion model [41], is another promising technique for visual synthesis. Therefore, we also evaluate the effectiveness of our method on diffusion model-based image and video generation with OmniTokenizer-VAE as the tokenizer. Here we employ the same architecture of DiT [34] and Latte [29] and replace their VAE [1] with OmniTokenizer-VAE. DiT [34] first applies the transformer architecture to latent diffusion models and exhibits appealing scalability properties. Following this, Latte [29] extends the transformer to the latent video diffusion model by alternating spatial and temporal attention blocks.

The experimental results, as depicted in Table 5, indicate that when equipped with OmniTokenizer-VAE, DiT-XL/2 with classifier-free guidance (CFG) achieves competitive results in terms of FID and inception score, underscoring the efficacy of our tokenizer within diffusion model frameworks for image synthesis. For unconditional video generation on the UCF-101 dataset, our method not only offers the advantage of reduced training costs by realizing a higher compression rate, but also exhibits a much lower FVD than previous methods.

### 4.4 Ablation Study

**Training Paradigms.** To verify the effect of the proposed progressive training paradigm, we compare different training strategies and show the results in Table 7. The results in lines 3-4 and line 6 indicate that joint training outperforms video training on all video datasets remarkably, demonstrating the importance of image pre-training for the following video training. In addition, although joint training on a fixed resolution (line 5) could achieve much better results on video datasets than video training, the reconstruction FID on ImageNet gets worse, *i.e.*, from 1.28 to 1.35. Comparatively, the progressive training paradigm leads to the best performance on video datasets and surprisingly improves the image reconstruction performance.

**Deeper analysis on attention mechanisms.** We conduct experiments on both image and video reconstruction using different attention variants to fully analyze its effect. As can be seen in Table 8, window attention outperforms plain attention for image reconstruction (ImageNet) since it incorporates the local modeling. While for video reconstruction (K600), using causal or not has little influence on the performance (Table 9), but causal attention is necessary since we train the following image or video generation transformer in an autoregressive manner.

Table 7: Comparison of rFID on ImageNet and rFVD on various video datasets.

| | Method | ImageNet | K600 | | UCF | | MiT | | SSV2 | |
|---|---|---|---|---|---|---|---|---|---|---|
| | | 256 | 128 | 256 | 128 | 256 | 128 | 256 | 128 | 256 |
| 1 | Ours-Image (Fix) | 1.28 | - | - | - | - | - | - | - | - |
| 2 | Ours-Image (Multi) | 1.44 | - | - | - | - | - | - | - | - |
| 3 | Ours-Video (Fix) | - | 211.51 | 48.89 | 214.83 | 118.52 | 211.07 | 64.47 | 162.53 | 22.82 |
| 4 | Ours-Video (Multi) | - | 194.51 | 54.89 | 211.83 | 114.52 | 238.07 | 26.47 | 193.35 | 38.82 |
| 5 | Ours-Joint (Fix) | 1.35 | 113.51 | 26.89 | 186.83 | 62.52 | 140.07 | 21.47 | 108.35 | 20.82 |
| 6 | Ours-Joint (Multi) | **1.11** | **84.38** | **25.97** | **107.80** | **42.35** | **59.47** | **19.87** | **84.78** | **20.30** |

Table 8: Spatial atten.

| Spatial Attention | rFID |
|---|---|
| Plain | 1.55 |
| Window | 1.28 |

Table 9: Temporal atten.

| Spatial | Temporal | rFVD |
|---|---|---|
| Window | Plain | 26.43 |
| Window | Causal | 25.97 |

Table 10: Ablation on the training order.

| Training | ImageNet | | K600 | |
|---|---|---|---|---|
| | VQVAE | VAE | VQVAE | VAE |
| Joint (VQ-KL) | 1.11 | 0.69 | 25.97 | 13.02 |
| Joint (KL-VQ) | 2.05 | 0.89 | 33.79 | 12.43 |

**Effects of training recipe.** We change the training recipe by performing KL training first and then VQ training, and the results in Table 10 show that this will lead to performance drop on both ImageNet and K600. We believe this is because going from discrete tokens to continuous embedding will make the learning of the model easier and more stable.

### 4.5 Visualizations

**Visual Reconstruction.** We visualize the reconstruction results by OmniTokenizer, VQGAN [12] and TATS [13] in Figure 3. Our method works significantly better than baselines for face and text reconstruction, which are typically regarded as the most challenging reconstruction cases.

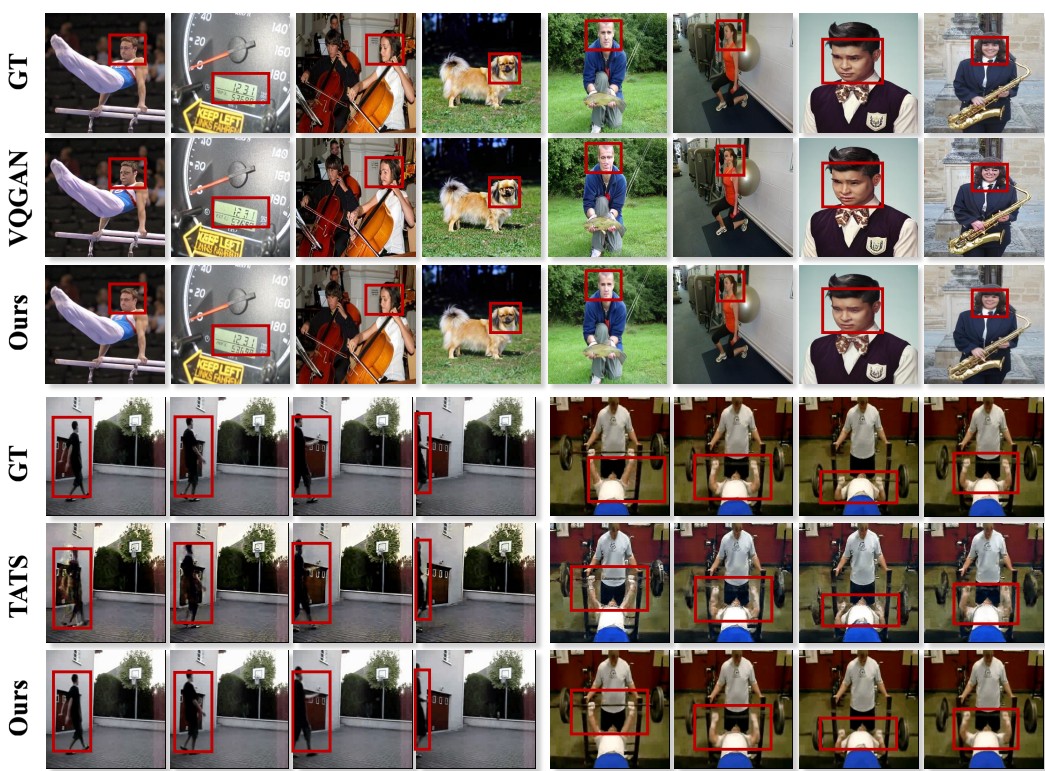

Figure 3: Image and video reconstruction results of VQGAN [12], TATS [13], and our method.

**Class-conditional Image and Video Generation.** The class-conditional generation results are shown in Figure 4-7. Our model could synthesize visually coherent and contextually accurate images and videos, showcasing the strengths of OmniTokenizer in facilitating generative tasks.

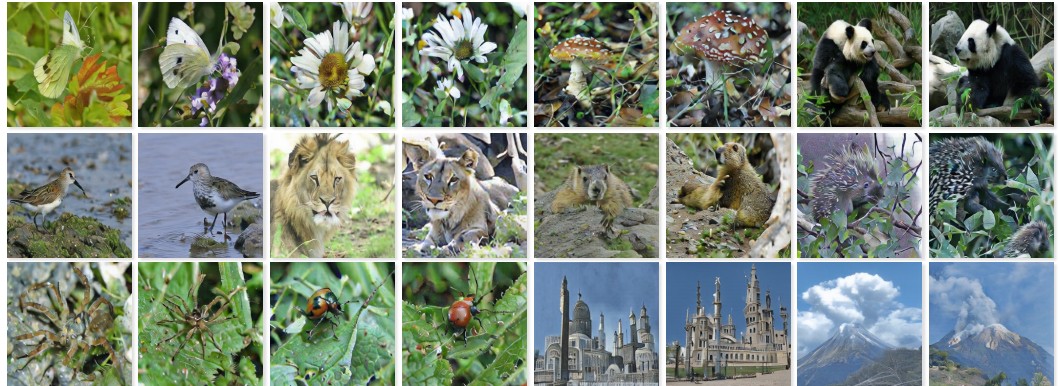

Figure 4: Class-conditional ImageNet generation results using language models, with OmniTokenizer-VQVAE as tokenizer.

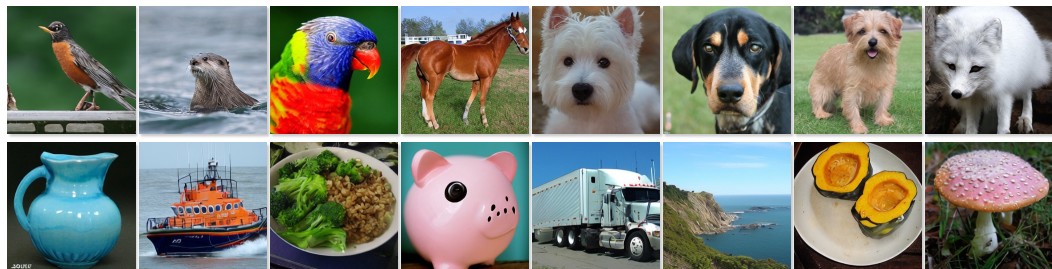

Figure 5: Class-cond. ImageNet generation using diffusion models, OmniTokenizer-VAE as tokenizer.

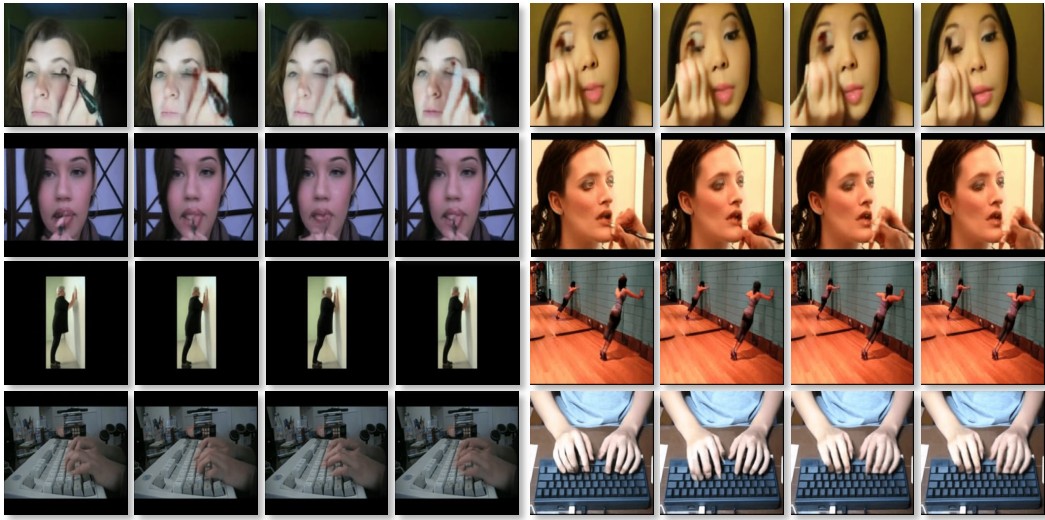

Figure 6: Class-cond. UCF-101 generation using LMs, OmniTokenizer-VQVAE as tokenizer.

**Frame Prediction and Arbitrary Long Video Generation.** The frame prediction results by our method are presented in Figure 8, from which we can see that our model could forecast subsequent frames with high clarity and temporal coherence. Moreover, we exhibit the potential of our method for generating videos of arbitrary lengths by employing a cyclical process, where each newly generated frame is recursively used as a condition for the subsequent frame generation.

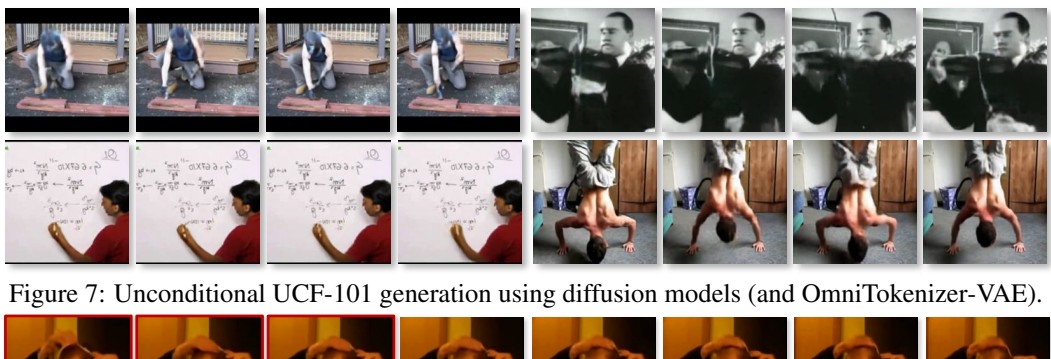

Figure 7: Unconditional UCF-101 generation using diffusion models (and OmniTokenizer-VAE).

Figure 8: Visualization of the frame prediction results by OmniTokenizer. The frames marked in red are given during inference, while the following frames are generated.

## 5 Conclusion and Discussion of Broader Impact

This paper presented OmniTokenizer, a transformer-based tokenizer for joint image-video tokenization. OmniTokenizer adopts a spatial-temporal decoupled architecture, employing the window and causal attention in the spatial and temporal dimensions. To realize the synergy between images and video data, we proposed a progressive training strategy that starts with image training on a fixed resolution to acquire the spatial encoding capability and then incorporates video data for multi-resolution joint training to learn temporal modeling. Extensive experimental results substantiate the state-of-the-art performance of OmniTokenizer in visual reconstruction tasks. Further, when equipped with OmniTokenizer, both language model-based methods and diffusion models could achieve superior visual generation results.

Previous literature [20, 16, 70, 48, 47] has revealed that the performance of transformer models improves significantly as the model size increases, also known as scaling law. In the future, we will explore scaling the model capacity of OmniTokenizer for more advanced tokenization performance.

**Acknowledgements.** This work was supported in part by National Natural Science Foundation of China (#62032006).

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
