# OpenReview forum: "OmniTokenizer: A Joint Image-Video Tokenizer for Visual Generation"
_NeurIPS.cc/2024/Conference — NeurIPS 2024 poster_

### Official Review · Reviewer_ZuXX · 2024-06-19

**Soundness:** 3
**Presentation:** 2
**Contribution:** 2
**Rating:** 6
**Confidence:** 5

**Summary:**

This paper proposes OmniTokenizer, a tokenizer that can be used to tokenize both image and video data. To train the OmniTokenizer, they devise an architecture that decouples spatial and temporal axis, offering efficiency. Next, they perform a progressive training scheme that initializes from image-only training at fixed resolution that is followed by a joint image & video training at multiple resolutions. The resulting VQ tokenizer can be used with autoregressive modeling objective to perform generation. Optionally, it can also be fine-tuned with KL loss to be used as a diffusion tokenizer for latent diffusion models. The evaluations are performed on several image and video datasets to demonstrate the effectiveness of the tokenizer.

**Strengths:**

Having a joint tokenizer for image and video modalities is an important step. From the results, the paper seems to do a decent job at it and the resulting tokenizers could be useful to the community. The ideas of decoupling spatial & temporal axis and progressive training are intuitive and seem effective from the provided ablation results. The paper is clearly written.

**Weaknesses:**

My main weakness point with the paper is the lack of comparisons to the relevant baselines. I think MagVITv2 is the closest baseline to your method, but there is no in-depth comparison to it. I can see in Table 4 that MagVITv2 was reported in NAR setting while yours is in AR setting. I think it's important to have an apples-to-apples comparison to understand the effectiveness of your method. Same applies to image generation results in Table 3. For the comparison, I wouldn't limit the comparison to only generation results but also other relevant & important aspects such as video compression and the quality of learned representations.

Other important and missing baselines are MAGE and MaskGIT. There are also recent advancements in the literature such as VAR but I wouldn't penalize for them as they are quite recent. Still, having a discussion & comparison to them would be useful.

Other questions and comments:
- How the trends would look like for 512x512 resolution?
- For the reconstruction results, having other metrics such as PSNR, SSIM, and IS would be helpful.
- Seeing the qualitative video results on actual videos would be really helpful. From the provided visuals it's unclear.
- Are you planning to release the checkpoints & code for OmniTokenizer?
- How do you explain the small performance gap for generation results with diffusion models? Can we say that OmniTokenizer is more effective with an autoregressive Transformer?
- It was mentioned that the decoder is symmetric with encoder. Could you clarify that and provide a full architecture visualization? Could you also clarify the 2D and 3D patch embeddings?
- What would be the reconstruction results for diffusion models on Table 1 & 2? You reported SD1.4 in the appendix. There are some numerical differences though (e.g. yours rFID is 0.69 in the main paper and 0.68 in the appendix), what is the reason for this? What are the reconstruction results for other latent diffusion models like SD3?
- In the ablation results (Table 7) what would be the performance if you skip the fixed resolution image training stage and perform only the second stage on image & video jointly?

**Questions:**

Please see the weaknesses for details. In particular, I am wondering an in-depth comparison against relevant baselines e.g. MagVITv2, MAGE, and MaskGIT. I am also wondering qualitative results on videos and some training & experimental clarifications I discussed in the weaknesses. I will update my final score accordingly.

**Limitations:**

Yes.

---

> ### Author Rebuttal · Authors · 2024-08-07
>
> 1. My main weakness point with the paper is the lack of comparisons to the relevant baselines. I think MagVITv2 is the closest baseline to your method, but there is no in-depth comparison to it. I can see in Table 4 that MagVITv2 was reported in NAR setting while yours is in AR setting. I think it's important to have an apples-to-apples comparison to understand the effectiveness of your method. Same applies to image generation results in Table 3. For the comparison, I wouldn't limit the comparison to only generation results but also other relevant & important aspects such as video compression and the quality of learned representations.
>
>    =>Answer: Please refer to question 2 of our global rebuttal.
>
> 2. Other important and missing baselines are MAGE and MaskGIT. There are also recent advancements in the literature such as VAR but I wouldn't penalize for them as they are quite recent. Still, having a discussion & comparison to them would be useful.
>
>    =>Answer: Thanks for your suggestion. We will add more discussion on MAGE, MaskGIT, and VAR in the main paper.
>
>
> 3. How the trends would look like for 512x512 resolution?
>
>    =>Answer: Good point! We evaluate VQGAN and our model at 512x512 resolution, as shown in Table 13 of rebuttal PDF, both rFID and rFVD get lower and our method still outperforms VQGAN.
>
>
>     | **Method** | **ImageNet-256** | **ImageNet-256** |  **UCF-256** |  **UCF-512** |
>     |:--------|:--------------:|:--------------:|:--------------:|:--------------:|
>     | VQGAN | 1.49 | 0.82 | 94.49 | 32.56 |
>     | Ours (VQVAE) | 1.11 | 0.72 | 25.97 | 14.09 |
>
> 4. For the reconstruction results, having other metrics such as PSNR, SSIM, and IS would be helpful.
>
>    =>Answer: Thanks for pointing this out. We evaluate the PSNR and SSIM of VQGAN and our method, and show the results in Table 14 (rebuttal PDF). On both image and video benchmarks, we surpass VQGAN.
>
>     | **Method** | **ImageNet-PSNR** | **ImageNet-SSIM** |  **UCF-PSNR** |  **UCF-SSIM** |
>     |:--------|:--------------:|:--------------:|:--------------:|:--------------:|
>     | VQGAN | 23.27 | 0.69 | 26.53 | 0.81 |
>     | Ours (VQVAE) | 24.96 | 0.77 | 28.89 | 0.90 |
>
> 5. Seeing the qualitative video results on actual videos would be really helpful. From the provided visuals it's unclear.
>
>    =>Answer: Great suggestion! Due to the rebuttal policy, we cannot update the supplementary to provide qualitative video results on actual videos, but we have shown more qualitative results in Figure 11 and 12 of our rebuttal PDF.
>
>
> 6. Are you planning to release the checkpoints & code for OmniTokenizer?
>
>    =>Answer: Yes, we will definitely open-source the checkpoints & code for OmniTokenizer.
>
>
> 7. How do you explain the small performance gap for generation results with diffusion models? Can we say that OmniTokenizer is more effective with an autoregressive Transformer?
>
>    =>Answer: Good question. For the experiments on diffusion models, we adopt the same hyper-parameters with DiT and Latte, to demonstrate the effects of replacing SD VAE with our VAE on the image/video generation performance. Therefore, these hyper-parameters are tuned for the original VAE, and we find that after tuning the learning rate for our model, the performance could be boosted, as shown in Table 15 of the rebuttal PDF (due to time limit, we only conduct experiments on Latte). Therefore, we believe OmniTokenizer is effective with both transformer models and diffusion models.
>
>     |**Method**|**Learning rate**|**FVD**|
>     |:--------|:--------------:|:--------------:|
>     |Latte| 1e-4 | 478.0 |
>     |Ours-Latte | 1e-4 | 525.6 |
>     |Ours-Latte | 2.5e-4 | 209.2 |
>
>
> 8. It was mentioned that the decoder is symmetric with encoder. Could you clarify that and provide a full architecture visualization? Could you also clarify the 2D and 3D patch embeddings?
>
>    =>Answer: Thanks for the suggestion. The detailed architecture is illustrated in Figure 13 (rebuttal PDF).
>
>
> 9. What would be the reconstruction results for diffusion models on Table 1 & 2? You reported SD1.4 in the appendix. There are some numerical differences though (e.g. yours rFID is 0.69 in the main paper and 0.68 in the appendix), what is the reason for this? What are the reconstruction results for other latent diffusion models like SD3?
>
>    =>Answer: Thanks for pointing this out. The rFID of our VAE model is 0.69 and sorry for the confusion. In Table 16 (rebuttal PDF), we compare the reconstruction performance of SD1.4 VAE and SD3 VAE with our tokenizer on ImageNet, K600, and UCF (Table 1&2). It is worth mentioning that although SD VAEs are trained on large-scale high-quality data, we still beat them on ImageNet and achieve on-par results on K600 and UCF using the publicly available academic datasets.
>
>     |**Method**|**ImageNet**|**K600**| **UCF**|
>     |:--------|:--------------:|:--------------:|:--------------:|
>     |SD1.4 VAE | 0.77 | 8.89 | 20.91 |
>     | SD3 VAE | 0.74 | 6.28 | 19.43 |
>     | Ours (VAE) | 0.69 | 13.02 | 23.44 |
>
> 10. In the ablation results (Table 7) what would be the performance if you skip the fixed resolution image training stage and perform only the second stage on image & video jointly?
>
>     =>Answer: Great Suggestion. We perform joint training without image pretraining and compared the results in Table 17 (rebuttal PDF). It can be seen that both ImageNet and K600 reconstruction performance get dropped compared to the progressive training, which are even worse than the image-only and video-only training. This shows the importance of image pre-training for video reconstruction learning, and also indicates that simple joint training could not improve the results of image reconstruction and video reconstruction.
>
>     |**Method**|**ImageNet**|**K600**|
>     |:--------|--------------|---------|
>     |Ours-Image (Fix)|1.28|-|
>     |Ours-Video (Multi)|-|54.89|
>     |Joint w/o Image (Multi)|2.62|67.77|
>     |Ours-Joint (Multi) |1.11|25.97|

---

> > ### Comment · Reviewer_ZuXX · 2024-08-12
> > **Thanks, rebuttal was helpful**
> >
> > Thank you for the detailed response and the new results you provided, they were helpful. Please incorporate all of them to your paper. I still think having an apples-to-apples comparison with MagVITv2 on visual generation will be useful to the practitioners to understand the capabilities of two approaches. Consider adding this to your paper as well.
> >
> > I'm raising my score to weak accept.

---

> > > ### Author Response · Authors · 2024-08-12
> > > **Thanks for your positive comments!**
> > >
> > > Thanks for your positive comments and we will follow your suggestion to add these results to our paper!

---

### Official Review · Reviewer_hbMY · 2024-07-10

**Soundness:** 3
**Presentation:** 2
**Contribution:** 3
**Rating:** 6
**Confidence:** 4

**Summary:**

This paper proposes OmniTokenizer, a transformer-based visual tokenizer model that processes both image and video input and achieves state-of-the-art reconstruction quality. OmniTokenizer's core designs are a decoupled spatial-temporal attention mechanism and a progressive training schedule. Two OmniTokenizers are trained for autoregressive generative models and diffusion models, respectively. Extensive experiments further show the new tokenizers effectively improve image and video generation performance, consolidating OmniTokenizer's effectiveness.

**Strengths:**

1. Visual tokenizers are essential components for representation learning and generation, laying the foundation for many cutting-edge techniques, including image/video diffusion models and multimodal large language models. This paper's efforts in improving the tokenization quality and reconstruction performance of visual tokenizers are appreciated.

2. The proposed OmniTokenizer employs image and video data jointly with a properly designed two-stage training paradigm. Ablation studies show that the joint two-stage training strategy effectively improves the reconstruction of both image and video data.

3. Two variants of OmniTokenizers are trained, one with VQ and one with the vanilla VAE's KL regularization. The paper further trains autoregressive and diffusion-based image/video generative models with the OmniTokenizers, demonstrating superior results across several image and video datasets.

**Weaknesses:**

1. Although there are extensive quantitative results on image/video reconstruction and generation, the qualitative comparisons are insufficient. Much more video reconstruction/generation comparisons can be provided in the supplementary materials to better demonstrate the effectiveness of OmniTokenizers.

2. The paper does not elaborate on the intuitions for designing decoupled window attention and causal attention. Table 8 investigates different architectural design choices, but there are no analyses on why the proposed design leads to better reconstruction quality than the other two variants.

3. Section 3.1.1 of the paper is too sketchy. To make the architecture reproducible and the paper self-contained, at least some concrete description or illustration of the window/causal attention mechanisms should be provided. Similarly, there are not many details about the image/video generative models trained with OmniTokenizers— the concrete model architecture/specifications and training setups could be provided in the appendix.

4. How is the order of VQ and KL training determined? There is no sufficient explanation on why the KL fine-tuning is conducted after the VQ training, and it's also unclear if the two variants of OmniTokenizers use two separate decoders or share the same decoder.

**Questions:**

The paper studies visual tokenizers and proposes an effective architecture with a joint image-video training framework. The thorough quantitative results convincingly demonstrate the advantages of the proposed OmniTokenizers, and further scaling up this framework could lead to more powerful tokenizers that benefit future research. However, the paper lacks several important information: i) qualitative comparisons (especially for videos), ii) details of the tokenizer architecture and insights on why it works better than other variants, and iii)  details of the image/video generative models. The lack of these details/analyses hurts the clarity and reproducibility. I will increase the score if these concerns are properly addressed in the rebuttal.

**Limitations:**

The limitations and broader impacts are discussed in Sections 4 and 5 of the main paper.

---

> ### Author Rebuttal · Authors · 2024-08-07
>
> 1. Although there are extensive quantitative results on image/video reconstruction and generation, the qualitative comparisons are insufficient. Much more video reconstruction/generation comparisons can be provided in the supplementary materials to better demonstrate the effectiveness of OmniTokenizers.
>
>    =>Answer: Great suggestion! Due to the rebuttal policy, we cannot update the supplementary, but we have shown more qualitative results in Figure 11 and 12 of our rebuttal PDF.
>
>
> 2. The paper does not elaborate on the intuitions for designing decoupled window attention and causal attention. Table 8 investigates different architectural design choices, but there are no analyses on why the proposed design leads to better reconstruction quality than the other two variants.
>
>    =>Answer: Please refer to question 3 of our global rebuttal.
>
>
> 3. Section 3.1.1 of the paper is too sketchy. To make the architecture reproducible and the paper self-contained, at least some concrete description or illustration of the window/causal attention mechanisms should be provided. Similarly, there are not many details about the image/video generative models trained with OmniTokenizers— the concrete model architecture/specifications and training setups could be provided in the appendix.
>
>    =>Answer: Thanks for your suggestion. We illustrate the network architecture in detail in Figure 13 (rebuttal PDF). The image and video transformers both follow a GPT-style architecture, which consist of 12 transformer layers. The number of heads is 12, hidden dimension is 768, and no dropout is used. We will add this introduction to our appendix.
>
>
> 4. How is the order of VQ and KL training determined? There is no sufficient explanation on why the KL fine-tuning is conducted after the VQ training, and it's also unclear if the two variants of OmniTokenizers use two separate decoders or share the same decoder.
>
>    =>Answer: Great question. OmniTokenizer-VQVAE and OmniTokenizer-VAE use two separate decoders and sorry for the confusion. We chose to perform VQ training before KL training because going from discrete tokens to continuous embedding is a process of information gain and vice versa is a process of information loss. Therefore, the current order will make the learning of the model easier and more stable. As shown in the Table 12 of rebuttal PDF, if KL training is performed before VQ training, the results of the latter will instead decrease. We will update this result in the main paper.
>
>     | **Training** | **ImageNet-VQVAE** | **ImageNet-VAE** |  **K600-VQVAE** |  **K600VAE** |
>     |:--------|:--------------:|:--------------:|:--------------:|:--------------:|
>     |Joint (VQ-KL) | 1.11 | 0.69 | 25.97 | 13.02 |
>     | Joint (KL-VQ) | 2.05 | 0.89 | 33.79 | 12.43 |

---

> > ### Comment · Reviewer_hbMY · 2024-08-13
> >
> > Thank you for providing the new results and more details of the model architecture. The intuitive explanation of the order of VQ and KL training makes sense to me, and is well backed up by the new ablation study. All my questions are resolved, and I will increase my rating to 6. If the paper is accepted, please incorporate the additional qualitative/quantitative results and the detailed architecture explanation into the paper (most of them can go into the appendix with a reference in the main paper).

---

> > > ### Author Response · Authors · 2024-08-13
> > > **Thanks for your positive comments!**
> > >
> > > Thanks for your valuable suggestions and positive feedback. We will incorporate these experimental results and the architecture explanation into the paper.

---

### Official Review · Reviewer_Nxg3 · 2024-07-12

**Soundness:** 3
**Presentation:** 2
**Contribution:** 2
**Rating:** 4
**Confidence:** 3

**Summary:**

The paper introduces OmniTokenizer, a transformer-based tokenizer designed for both image and video tokenization within a unified framework. This tokenizer employs a spatial-temporal decoupled architecture, using window attention for spatial and causal attention for temporal modeling. The approach leverages a progressive training strategy, starting with image data to develop spatial encoding skills and then extending to video data to handle temporal dynamics. Extensive testing on several datasets demonstrates that OmniTokenizer achieves good performance in reconstruction tasks and enhances the effectiveness of both AR-based and diffusion based generative models.

**Strengths:**

- The progressive training strategy is intuitive, allowing the model to learn incrementally from simpler to more complex tasks. This strategy contributes to the model's performance and makes sense to me.
- The KL fine-tuning for converting a VQ tokenizer to a continuous tokenizer seems interesting and few works explored that.

**Weaknesses:**

- The model does not compare with magvitv2 thoroughly, which should be a strong baseline in terms of reconstruction and generation.
- The architecture for single frame mostly follows ViT-VQGAN and the improvement comes from joint ImageNet/UCF training, when compared with single frame baselines in Table 1/3/4 the comparison seems unfair as the baseline tokenizers are trained on ImageNet only, but the UCF data can also be used as a single frame data source.
- The factorized spatial-temporal attention is not novel as many previous works also employed that technique.

**Questions:**

Overall I believe this paper shows some interesting points like progressive training and KL fine-tuning. However, the unfair comparison with baselines (magvitv2 and others) prevents me from giving higher ratings, and it is difficult to evaluate the significance of the performance given the current results

---

> ### Author Rebuttal · Authors · 2024-08-07
>
> 1. The model does not compare with magvitv2 thoroughly, which should be a strong baseline in terms of reconstruction and generation.
>
>     =>Answer: Please refer to question 2  of our global rebuttal.
>
>
> 2. The architecture for single frame mostly follows ViT-VQGAN and the improvement comes from joint ImageNet/UCF training, when compared with single frame baselines in Table 1/3/4 the comparison seems unfair as the baseline tokenizers are trained on ImageNet only, but the UCF data can also be used as a single frame data source.
>
>     =>Answer: Good question. Our goal is to enable image and video tokenization in a unified framework and achieve mutual benefits between them. To achieve this, it is necessary and natural to use both image and video data to train our model. In addition, as shown in the Table 17(rebuttal PDF), simply training on image and video data cannot lead to performance gains on image and video reconstruction, highlighting the importance of the proposed training strategy.
>
>     |**Method**|**ImageNet**|**K600**|
>     |:--------|:--------------:|:---------:|
>     |Ours-Image (Fix)|1.28|-|
>     |Ours-Video (Multi)|-|54.89|
>     |Joint w/o Image (Multi)|2.62|67.77|
>     |Ours-Joint (Multi) |1.11|25.97|
>
>
> 3. The factorized spatial-temporal attention is not novel as many previous works also employed that technique.
>
>     =>Answer: Please refer to question 1 of our global rebuttal.

---

> ### Author Response · Authors · 2024-08-12
> **Help check whether questions are well answered.**
>
> Dear reviewer Nxg3,
>
> We would like to thank you again for your effort and valuable suggestions. Can you help find time to take a look at the response and check whether your questions are well answered. We are very happy to discuss with you and provide further clarification for any new question.

---

> > ### Comment · Reviewer_Nxg3 · 2024-08-13
> >
> > I have read the authors' responses and other reviewers' comments. I appreciate the authors' efforts on providing additional results and choose to raise my rating to weak accept.

---

> > > ### Author Response · Authors · 2024-08-13
> > > **Thanks for raising the rating to weak accept**
> > >
> > > Dear reviewer Nxg3, thanks for your effort in reviewing our paper and choosing to raise your rating to weak accept!

---

### Official Review · Reviewer_pXcq · 2024-07-20

**Soundness:** 3
**Presentation:** 3
**Contribution:** 3
**Rating:** 6
**Confidence:** 5

**Summary:**

The paper introduces OmniTokenizer, a transformer-based image-video tokenizer designed for visual generation tasks. It adopts a spatial-temporal decoupled architecture, integrating window attention for spatial modeling and causal attention for temporal dynamics, allowing it to process both image and video data within a unified framework. A progressive training strategy is proposed, starting with image data to develop spatial encoding and then incorporating video data for learning temporal features across multiple resolutions. Extensive experiments demonstrate OmniTokenizer's state-of-the-art performance in reconstruction tasks on various datasets, outperforming previous methods.

**Strengths:**

1. The proposed tokenizer can enhance the performance of both language model-based and diffusion model-based visual generation approaches.
2. The key of this work lies in its potential to provide a versatile (image and video) and efficient tool for translating complex visual data into compact latent representations.
3. This paper is generally well-written and clearly stated.

**Weaknesses:**

1. The novelty is limited. The proposed tokenizer architecture is not new and is basically the same as previous methods (e.g., MagVit). The only difference is further regarding images as a 1-frame video to pretrain the network. I don’t think that counts as enough contributions.
2. The spatial-temporal decoupled architecture is a key aspect of OmniTokenizer. A deeper dive into the role and impact of different attention mechanisms on the model's performance could offer more clarity on design choices and potential improvements.
3. While the paper mentions the potential for scaling up the model capacity, it does not provide a detailed analysis of how the model scales with increased data volume or model size. Providing insights into scalability, such as computational complexity, training time, and resource requirements, would be valuable.

**Questions:**

See weakness

**Limitations:**

Yes.

---

> ### Author Rebuttal · Authors · 2024-08-07
>
> 1. The novelty is limited.
>
>     =>Answer: Please refer to question 1 of our global rebuttal.
>
>
> 2. The spatial-temporal decoupled architecture is a key aspect of OmniTokenizer. A deeper dive into the role and impact of different attention mechanisms on the model's performance could offer more clarity on design choices and potential improvements.
>
>     =>Answer: Please refer to question 3 of our global rebuttal.
>
>
> 3. While the paper mentions the potential for scaling up the model capacity, it does not provide a detailed analysis of how the model scales with increased data volume or model size. Providing insights into scalability, such as computational complexity, training time, and resource requirements, would be valuable.
>
>     =>Answer: Great suggestion. We leave the model scaling as our future work, as previous literature indicates that transformer models always exhibit promising scalability. We will study how the model scales with increased data or model size when more resources are available.

---

### Author Rebuttal · Authors · 2024-08-07

We would like to thank all reviewers for their valuable comments. We are happy the reviewers think the progressive training strategy is **intuitive** [Review pXcq, Reviewer ZuXX] and **effective** [Review hbMY, Reviewer ZuXX]. Below we respond to the common concerns of reviewers.
1. Novelty of this work

    =>Answer: We would like to emphasize that our primary contribution is not an innovation in model architecture, but a novel progressive training strategy. It offers twofold significant advantages over existing methods. First of all, progressive training enables image and video tokenization with one model and one weight (casting the image and video inputs to the same codebook), which allows the joint training of image and video generation with one tokenizer. Secondly, compared to image-only and video-only training, the proposed progressive training could lead to better performance on both image and video reconstruction, as can be seen from Table 7 (main paper). This is not easy to achieve, since simply training an image tokenizer like ViTVQGAN on video data will hurt the performance on image reconstruction, as verified in  Table 7 (main paper) and Table 17 (rebuttal PDF).

    |**Method**|**ImageNet**|**K600**|
    |:--------|:--------------:|:---------:|
    |Ours-Image (Fix)|1.28|-|
    |Ours-Video (Multi)|-|54.89|
    |Joint w/o Image (Multi)|2.62|67.77|
    |Ours-Joint (Multi) |1.11|25.97|

2. Comparison with MAGVITv2

    =>Answer: We strongly agree that Magvitv2 is a strong baseline for our method. However, since it is not open-source, we can only look for the results in its paper and compare with it as thoroughly as possible.
    For visual reconstruction, as shown in Table 10 (left) of our rebuttal PDF, our method is better than Magvitv2 on ImageNet, while on UCF, we are worse than MagViTv2 but outperforms previous SOTAs like TATS significantly. For visual generation, we adopt different generative model architectures, i.e., autoregressive v.s. non-autoregressive transformers, so it is not a fair comparison. We believe that the reconstruction performance can better reflect the capability of  different tokenizers.


   |**Method**|**ImageNet**|**UCF**|
   |:--------|:--------------:|:---------:|
   |TATS|-|162|
   |MAGVITv2|1.15|9|
   |Ours-VQVAE |1.11|42|
   |Ours-VAE |0.69|23|

    We also compare with Magvitv2 on video understanding tasks, i.e., action recognition on K600. We follow them to use tokens as the input to the ViViT transformer. It can be seen from Table 10 (right) that our methods achieve on-par performance with MagViTv2 and outperforms MagViT by a clear margin.
   |**Method**|**K600**|
   |:--------|:--------------:|
   |3D VQ-VAE|45.67|
   |MAGVIT|74.65|
   |MAGVITv2|77.93|
   |Ours (VQVAE)|77.34|

    In addition, it is worth mentioning that our work differs from MAGVITv2 in that they need to train separate models for image and video tokenization, but the proposed progressive training strategy allows us to tokenize image and video inputs with the same model and weight, and to the best of our knowledge we are the first to achieve this.

3. A deeper dive into the attention mechanisms

    =>Answer: We do ablation studies on attention mechanisms in Table 8 (main paper). In order to fully analyze its effect, we additionally conduct experiments on both image and video reconstruction using different attention variants. As can be seen in Table 11 (rebuttal PDF), window attention outperforms plain attention for image reconstruction (ImageNet) since it incorporates the local modeling. While for video reconstruction (K600), using causal or not has little influence on the performance, but causal attention is necessary since we train the following image or video generation transformer in an autoregressive manner.

    |**Spatial**| **rFID**|
    |:--------|:--------------:|
    |Plain|1.55|
    |Window | 1.28 |


    |**Spatial**| **Temporal** | **rFVD**|
    |:--------|:--------------:|:--------------:|
    |Window|Plain| 26.43|
    |Window | Causal | 25.97|

---

### Author Response · Authors · 2024-08-11
**Additional Notes on MAGVITv2 Comparison**

MAGVITv2 utilizes an exceptionally large codebook size (2^18), whereas we have used only 16k, which precludes a fair comparison. Additionally, since MAGVITv2 is not open-source, reproducing its work to achieve a fair comparison is challenging. In future work, we will compare performance using the same codebook size.

---

### Decision · Program_Chairs · 2024-09-25

**Decision:**

Accept (poster)

**Comment:**

Reviewers initially had concerns about the paper, namely around the contribution and lack of comparison to magvitv2. In rebuttal authors justified that this baseline was not open source and would be difficult to reproduce for proper comparison. They also clarified the contribution and provided additional results requested by reviewers. All reviewers increased their scores to weak accept (one did not by lapse -- they told authors they would).

While the contribution is somewhat subtle (a "progressive training strategy" for learning a joint image-video tokenizer), all reviewers believe the results are strong enough that there may be something to it, and hence deserves acceptance.